# What are the risks and benefits of temporarily discontinuing medications to prevent acute kidney injury? A systematic review and meta-analysis

Penny Whiting,[1,2] Andrew Morden,[1,2] Laurie A Tomlinson,[3,4] Fergus Caskey,[2,3] Thomas Blakeman,[5,6] Charles Tomson,[7] Tracey Stone,[1,2] Alison Richards,[1,2] Jelena Savović,[1,2] Jeremy Horwood[1,2]

▶ Prepublication history and additional material is available. To view please visit the journal (http://dx.doi.org/10.1136/bmjopen-2016-012674).

For numbered affiliations see end of article.

**Correspondence to**
Dr Penny Whiting;
penny.whiting@bristol.ac.uk

## ABSTRACT

**Objectives:** To summarise evidence on temporary discontinuation of medications to prevent acute kidney injury (AKI).

**Design:** Systematic review and meta-analysis of randomised and non-randomised studies.

**Participants:** Adults taking diuretics, ACE inhibitors (ACEI), angiotensin receptor blockers (ARB), direct renin inhibitors, non-steroidal anti-inflammatories, metformin or sulfonylureas, experiencing intercurrent illnesses, radiological or surgical procedures.

**Interventions:** Temporary discontinuation of any of the medications of interest.

**Primary and secondary outcome measures:** Risk of AKI. Secondary outcome measures were estimated glomerular filtration rate and creatinine post-AKI, urea, systolic and diastolic blood pressure, death, clinical outcomes and biomarkers.

**Results:** 6 studies were included (1663 participants), 3 randomised controlled trials (RCTs) and 3 prospective cohort studies. The mean age ranged from 65 to 73 years, and the proportion of women ranged from 31% to 52%. All studies were in hospital settings; 5 evaluated discontinuation of medication prior to coronary angiography and 1 prior to cardiac surgery. 5 studies evaluated discontinuation of ACEI and ARBs and 1 small cohort study looked at discontinuation of non-steroidal anti-inflammatory drugs. No studies evaluated discontinuation of medication in the community following an acute intercurrent illness. There was an increased risk of AKI of around 15% in those in whom medication was continued compared with those in whom it was discontinued (relative risk (RR) 1.17, 95% CI 0.99 to 1.38; 5 studies). When only results from RCTs were pooled, the increase in risk was almost 50% (RR 1.48, 95% CI 0.84 to 2.60; 3 RCTs), but the CI was wider. There was no difference between groups for any secondary outcomes.

**Conclusions:** There is low-quality evidence that withdrawal of ACEI/ARBs prior to coronary angiography and cardiac surgery may reduce the incidence of AKI. There is no evidence of the impact of drug cessation interventions on AKI incidence during intercurrent illness in primary or secondary care.

## Strengths and limitations of this study

- We have conducted a thorough systematic review of the evidence from studies that have examined interventions involving temporary discontinuation of medications to prevent or minimise the severity, or consequences, of acute kidney injury (AKI).
- This is a topic of major importance due to interventions currently being implemented to reduce the risk of AKI throughout the UK and internationally.
- Broad eligibility criteria included randomised and non-randomised studies; primary and secondary care; intercurrent illness or a radiological/surgical procedure; planned and unplanned settings.
- The strength of the conclusion is limited by the quality and number of studies, and absence of evidence for important settings and classes of medications.

**Trial registration number:** PROSPERO CRD42015023210.

## BACKGROUND

Acute kidney injury (AKI) is a sudden decline in renal function, affecting up to 20% of people admitted to hospital, and is strongly associated with increased mortality and longer duration of hospital stay.[1] Historically, recognition and treatment of AKI has been poor.[2] Recent comprehensive initiatives in the UK have focused on improving awareness and treatment of people with or at risk of AKI.[3] It is thought that a substantial proportion of AKI is triggered or exacerbated by prescribed medications, particularly during times of physiological stress such as

intercurrent illness, surgery or radiocontrast imaging.[4] These medications include ACE inhibitors (ACEI), angiotensin receptor blockers (ARB), diuretics, nonsteroidal anti-inflammatory drugs (NSAIDs). Under the same circumstances, reduced excretion of metformin is associated with an increased risk of lactic acidosis, while sulfonylureas can lead to a greater incidence of hypoglycaemia. Therefore, many clinicians, expert consensus statements and guidelines recommend that some or all of these medications are stopped prior to elective or emergency procedures, or when patients become unwell with symptoms of severe infection.[5] [6] Initiatives advising patients prescribed these medications to temporarily stop taking them when they become unwell (so-called 'sick-day rules') have been implemented throughout Scotland and in local initiatives across the UK.[7] However, the evidence base to support these recommendations is unclear, and the overall benefit remains controversial.[8]

We conducted a systematic review and meta-analysis of the randomised and non-randomised studies that have examined temporary discontinuation of all or any of these medications in patients in primary or secondary care at risk of AKI or with newly diagnosed AKI as a result of an intercurrent illness or a radiological/surgical procedure (planned or unplanned).

## METHODS

Systematic review methods followed guidance from the Centre for Reviews and Dissemination (CRD)[9] and the Cochrane Collaboration;[10] this review is reported according to the PRISMA guidelines.[11] The review followed a predefined published protocol.[12]

### Study eligibility criteria

Studies, randomised and non-randomised, that evaluated adults (age ≥18 years) who were taking a specified medication and experiencing an intercurrent illness or undergoing a radiological/surgical procedure (planned or unplanned) in whom the medication was temporarily discontinued for any reason were eligible for inclusion. Medications of interest were diuretics, ACEIs, ARBs, direct renin inhibitors, NSAIDs, metformin or sulfonylureas. Studies had to report a measure of kidney function (eg, incidence of AKI, estimated glomerular filtration rate (eGFR) or serum creatinine) and include a comparator group consisting of placebo, no treatment or usual care.

### Identification and selection of studies

The following databases were searched from inception to January 2016: Embase, MEDLINE, PsycINFO, BIOSIS Citation Index (Web of Science), CINAHL (Cumulative Index to Nursing and Allied Health Literature), Science Citation Index (SCI) (Web of Science) and the Cochrane Central Register of Controlled Trials (CENTRAL). Supplementary searches were undertaken to identify grey literature, completed and ongoing trials, in the following resources: NIH ClinicalTrials.gov (http://www.clinicaltrials.gov), metaRegister of Controlled Trials (http://www.controlled-trials.com), WHO International Clinical Trials Registry Platform (ICTRP) (http://www.who.int/ictrp/en), relevant guidelines (eg, NICE in the UK) regarding management of AKI. Reference lists of included studies were screened. Details of the MEDLINE search strategy are available as an online supplementary appendix. Search results and full-text articles were independently assessed for inclusion by two reviewers; disagreements were resolved through consensus or referral to a third reviewer where necessary.

### Data extraction and assessment of risk of bias

We extracted data on baseline characteristics (number of participants, participant characteristics, study settings, study design, country, inclusion and exclusion criteria), intervention/exposure related to stopping medication and outcomes. The primary outcome was incidence of AKI. Secondary outcomes included urinary biomarkers, clinical outcomes, creatinine, eGFR, urea and blood pressure. For dichotomous data (eg, incidence of AKI), we extracted the number of events and participants in each treatment group and calculated the relative risk (RR) and 95% CI. For continuous data, we extracted the mean and SD in each treatment group and calculated mean differences (MD) and 95% CIs.

Randomised controlled trials (RCTs) were assessed for methodological quality using a draft version of the new Cochrane risk of bias tool[13] that includes items covering allocation bias (random sequence generation, allocation concealment and baseline imbalance), departures from interventions (participant and study personnel blinding, deviations from intended interventions and analysis in groups to which they were randomised), attrition bias (incomplete outcome data and robustness of results to missing data), detection bias (blinding of outcome assessors and likelihood of blinding to have influenced results) and reporting bias (selective reporting of outcome domain being assessed). The ROBINS-I tool was used to assess the risk of bias in non-randomised studies.[14] It includes domains covering bias due to confounding, bias in the selection of participants into the study, bias due to departures from intended interventions, bias due to missing data, bias in taking measurements and bias in the selection of the reported result.

Data were extracted by one reviewer using a standard data extraction form designed for this review, and checked by a second reviewer. The risk of bias assessment was performed independently by two reviewers. Any disagreements were resolved by consensus or referral to a third reviewer.

### Data synthesis

We grouped studies by design (randomised vs non-randomised), population (coronary angiography vs surgery) and outcome. If there were two or more studies assessing the same outcome, data were plotted on a

forest plot. If data were considered statistically and clinically sufficiently homogeneous, then summary estimates were produced using random effects meta-analysis. When the same outcomes were assessed in randomised and non-randomised studies that were considered similar in terms of population and intervention, we first stratified the analysis based on study design. If summary estimates from stratified analyses were considered sufficiently similar, we then produced an overall summary estimate combining data from randomised and non-randomised studies. For dichotomous outcomes, we estimated summary RRs and 95% CIs; for continuous data, we estimated summary MDs and 95% CIs. Heterogeneity was investigated using forest plots and the $I^2$ statistic. Where data were considered too heterogeneous to pool, a narrative synthesis was provided. We used GRADE to rate the overall quality of the evidence for risk of bias, publication bias, imprecision, inconsistency, indirectness and magnitude of effect.[15]

## RESULTS

### Search results

The searches identified 4316 hits (records) of which 42 were considered potentially relevant and obtained for full-text review (figure 1). A total of six studies (1663 participants) were included in the review: three RCTs (522 participants)[16–18] and three prospective cohort studies (1141 participants).[19–21] One study was available only as a conference abstract and so limited details were available for this study.[21]

All studies were conducted in hospital settings: five evaluated discontinuation of medication prior to coronary angiography and one prior to cardiac surgery.[19] All but one study[16] restricted inclusion to patients deemed at higher risk of AKI such as those with chronic kidney disease (3 studies),[17 18 20] diabetes (1 study)[21] or a set of criteria that defined patients at high risk (1 study).[19] The most commonly reported comorbidities included diabetes, hypertension and congestive heart failure. No studies of discontinuation of medication in the community following an acute intercurrent illness were found. Studies were conducted in North America, Turkey and Israel. The mean age, where reported, ranged from 65 to 73 years and the proportion of women in the studies ranged from 31% to 52%. Five studies evaluated discontinuation of ACEI and ARBs, one small cohort study looked at discontinuation of NSAIDs.[20] The time point at which the medication was stopped varied between studies. Three studies reported that medication was stopped 24 hours prior to the procedure,[16 18 21] two (including the one study of surgery) that it was stopped on the morning of the procedure[17 19] and one (the study of NSAIDs)[20] did not provide details on when

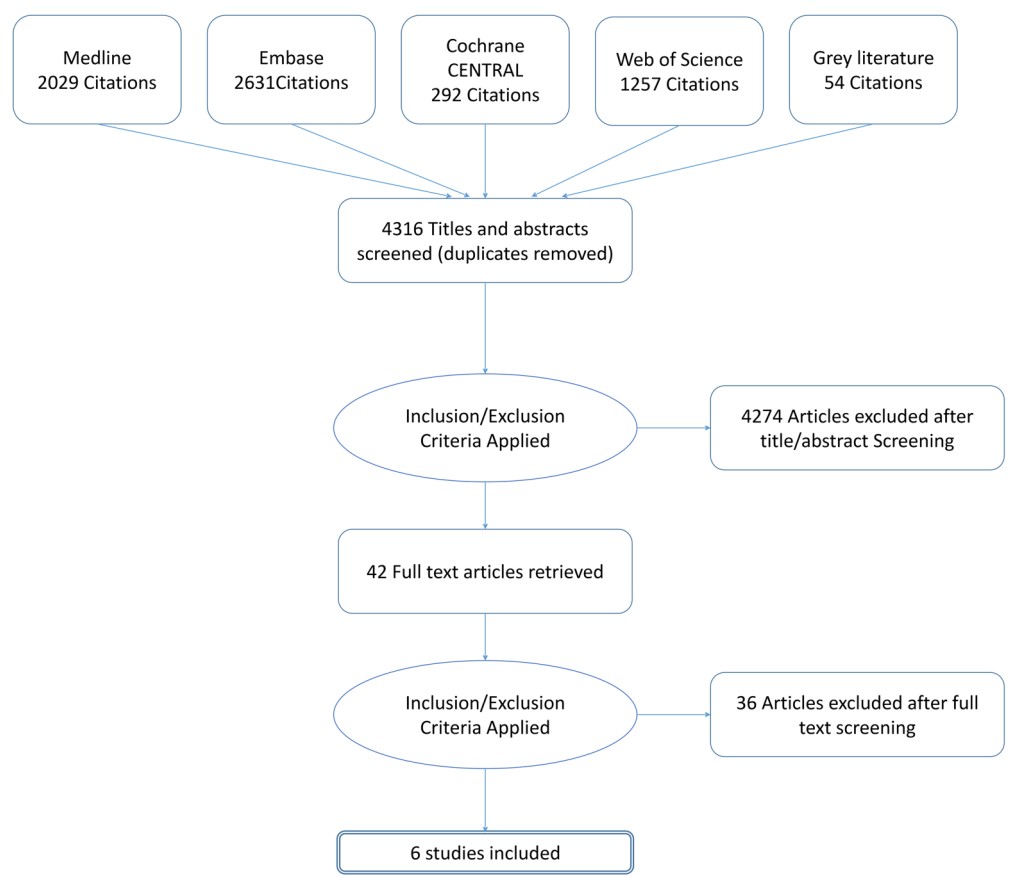

**Figure 1** Flow of studies through the review process.

medication was stopped. The time point at which medication was started again also varied. One study stated that medication was started up to 96 hours postprocedure,[18] one RCT[16] included two intervention arms that compared restarting immediately postprocedure with restarting 24 hours after the procedure and four studies did not report on this. No studies were found that assessed discontinuation of diuretics, metformin or sulfonylureas. Table 1 provides an overview of included studies.

## Risk of bias

The risk of bias assessment was performed for the primary outcome of incidence of AKI. Two RCTs were judged to have 'some concerns' regarding risk of bias[16 17] and one was rated 'low risk of bias'.[18] The two judged at 'some concern' had issues with the randomisation process; all other bias domains were rated 'low risk'. One provided no information on the methods used to allocate participants to the two treatment groups, the other randomised patients by means of a coin toss but

**Table 1** Details of studies included in the review

| | Bainey et al[18] | Rosenstock et al[17] | Wolak et al[16] | Coca et al[19] | Goksuluk et al[21]* | Weisbord et al[20] |
|---|---|---|---|---|---|---|
| Study design | RCT | RCT | RCT | Prospective cohort | Prospective cohort | Prospective cohort |
| Sample size | 208 | 220 | 94 | 1017 | 80 | 44 |
| Country | Canada | USA | Israel | North America | Turkey | USA |
| Population | Coronary angiography | Coronary angiography | Coronary angiography | Cardiac surgery | Coronary angiography | Coronary angiography |
| Risk group | CKD | CKD | None | High risk of AKI | Diabetes | CKD |
| Mean age (SD) | Intervention: 73 (9) Control: 72 (8) | Intervention: 72 (10) Control: 72 (10) | 65 (12) | Intervention: 71 (11) Control: 70 (12) | NR | NR |
| Female (%) | 26 | 52 | 33 | 31 | NR | NR |
| AKI definition | Increase in SCr ≥25% or ≥0.5 mg from baseline | Increase in SCr >25% or 0.5 mg from baseline | Increase in SCr ≥25% from baseline | Increase in SCr ≥50% or ≥0.3 mg from baseline | Increase in SCr ≥25% or ≥0.5 mg from baseline | Increase in SCr ≥25% from baseline or ≥0.5 mg from baseline |
| Comorbidities | Diabetes (54%), hypertension (47%), congestive heart failure (14%), liver cirrhosis (1%) | Hypotension (97%), diabetes (55%) | Diabetes (50%), unstable angina (62%) | Diabetes (47%), Hypertension (88%), congestive heart failure (23%) | Diabetes (100%) | NR |
| Study drug Intervention: timing of hold | ACE/ARB 24 hours prior to procedure | ACE/ARB Day of procedure | ACE/ARB 24 hours prior to procedure | ACE/ARB Morning of surgery | ACE/ARB 24 hours before procedure | NSAIDs No details |
| Intervention: timing of restart | Up to 96 hours postprocedure | 24 hours postprocedure | (1) Immediately afterwards; (2) 24 hours after | No details | No details | No details |
| Control | Continued throughout study | Continued throughout study | Continued throughout study | Continued throughout study | Continued throughout study | Continued throughout study |
| Risk of bias | Low | Some: randomised by coin toss, no information on allocation concealment. Baseline difference compatible with chance | Some; no information on treatment allocation, baseline difference compatible with chance | Moderate; controlled for confounding but possibility of residual confounding | Critical; no control for confounding | Not assessed |

*Available only as conference abstract.
ACE, angiotensin-converting enzyme inhibitors; AKI, acute kidney infection; ARB, angiotensin receptor blockers; CKD, chronic kidney disease; NSAIDs, non-steroidal anti-inflammatory drugs; RCT, randomised controlled trial; SCr, serum creatinine.

did not provide any information on whether allocation was concealed. Both studies provided a reasonable overview of baseline characteristics, including similarities in timings of baseline kidney function, which suggested that any differences between groups were compatible with chance. The risk of bias assessment highlighted that none of the studies provided information on blinding of participants, study personnel or outcome assessors. However, there do not appear to have been any departures from the intended interventions, thus knowledge of the assigned intervention appears unlikely to have influenced the study result. The outcome measure was considered relatively objective and therefore also unlikely to have been influenced by knowledge of treatment assignment.

It was not possible to conduct a risk of bias assessment for one of the non-randomised studies[20] as this study did not provide any numerical data and the risk of bias assessment is performed at the result level. One of the non-randomised studies was judged at moderate risk of bias,[19] and the other at critical risk of bias.[21] The study judged at critical risk of bias only presented crude outcome data with no adjustment for potential confounding factors. It was judged at low risk of bias for all other domains with the exception of measurement of interventions which was judged at moderate risk of bias as it was not clear exactly how exposure to ACEI and ARBs was measured. The study judged at moderate risk of bias was judged to have appropriately controlled for confounding factors, but the guidance for the ROBINS-I tool states that this domain can only be rated as low risk of bias if the study is considered comparable to a well-performed randomised trial.

## Incidence of AKI

One study did not provide any numerical data on the effect of discontinuation of medication on patient outcomes.[20] This cohort study, which assessed discontinuation

of NSAIDs, only found 3 patients out of 44 NSAID users who were advised to discontinue their medication prior to coronary angiography. It reported that discontinuation of NSAID was not associated with a lower rate of AKI, but this was limited by the small number of patients in whom medication was discontinued.

All other studies assessed the incidence of AKI (or contrast-induced nephropathy) in those in whom medication was stopped prior to the procedure compared with those in whom medication was continued (figure 2). Three studies defined AKI as an increase in creatinine of 25% or 0.5 mg/dL above baseline,[17 18 21] one as an increase in creatinine of 25% above baseline[16] and one used a slightly different definition of an increase in creatinine of 50% or 0.3 mg/dL above baseline.[19] All but one suggested an increased risk of AKI in those in whom medication was continued, but CIs were generally wide. There was an increased risk of AKI of around 15% in those in whom medication was continued compared with those in whom it was discontinued (RR 1.17, 95% CI 0.99 to 1.38). Omitting the study judged at critical risk of bias had very little effect on the summary estimate (RR 1.16, 95% CI 0.98 to 1.37). When only results from RCTs were pooled, the increase in risk was almost 50% (RR 1.48, 95% CI 0.84 to 2.60; 3 RCTs), but the CI was much wider. There was no evidence of heterogeneity for any of these analyses ($I^2=0\%$). Based on GRADE, the quality of the evidence was judged as low for the analysis restricted to RCTs and very low when non-randomised studies were included (table 2). The evidence was downgraded due to imprecision and the likelihood of publication bias for the analysis that included RCTs and for study quality and publication bias for the analysis that included non-randomised trials.

## Secondary outcomes

Two studies[16 17] assessed GFR and creatinine at 24 hours postintervention (figures 3 and 4). Both suggested no

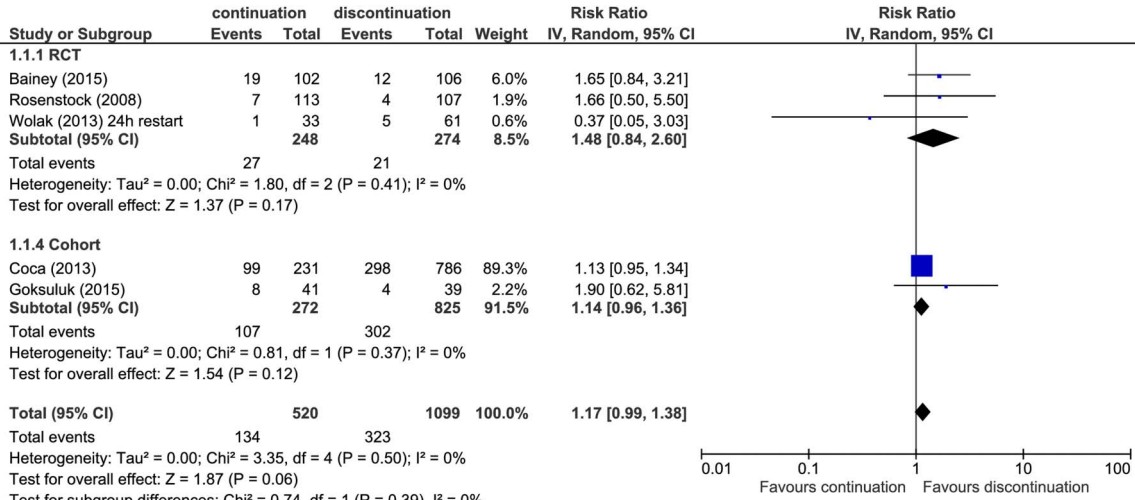

**Figure 2** Forest plot showing the risk of acute kidney injury in those who stopped medication prior to procedure compared with those who continued medication.

| Table 2 | GRADE evidence profile: risks and benefits of temporarily discontinuing medications to prevent acute kidney injury | | | | | | | | | |
|---|---|---|---|---|---|---|---|---|---|---|
| **Quality assessment** | | | | | | **Number of patients** | | **Effect** | | |
| Studies | Risk of bias | Inconsistency | Indirectness | Imprecision | Other considerations | Continuation | Discontinuation | Relative (95% CI) | Absolute (95% CI) | Quality |
| *Incidence of acute kidney injury* | | | | | | | | | | |
| 3 RCTs | Not serious | Not serious | Not serious | Serious* | Publication bias strongly suspected† | 27/248 (10.9%) | 21/274 (7.7%) | RR 1.48 (0.84 to 2.60) | 52 more per 1000 (from 17 fewer to 174 more) | ⊕⊕◯◯ Low |
| 3 RCTs3 Cohorts | Very serious‡ | Not serious | Not serious | Not serious | Publication bias strongly suspected† | 134/520 (25.8%) | 323/1099 (29.4%) | RR 1.14 (0.96 to 1.36) | 36 more per 1000 (from 10 fewer to 93 more) | ⊕◯◯◯ Very low |

*Wide CI and few events.
†Non-randomised studies appear would have been unlikely to have been written up for publication if findings had been negative; therefore, similar studies with negative findings considered likely.
‡RCTS, no serious concerns regarding risk of bias. Two cohort studies, one judged moderate risk of bias due to possibility of residual confound, one judged critical risk of bias as did not control for confounding.
RR, Risk ratio.

difference in these measures between intervention groups, although CIs were wide. Other outcomes reported in single studies included urinary biomarkers (structural AKI), clinical end points (mortality, myocardial infarction, stroke, congestive heart failure, rehospitalisation, hypertensive treatment) and blood pressure. Generally, there was no difference between groups in which medication was stopped and groups in which it was continued for any of these outcomes. Table 3 provides an overview of key outcomes; other outcomes reported in included studies were different ways of measuring these outcomes (eg, continuous rather than dichotomous data, or change from baseline rather than absolute value).

## DISCUSSION

The results of our meta-analysis demonstrate a ~15% increased risk of AKI in those in whom medication was continued compared with those in whom it was discontinued (RR 1.17, 95% CI 0.99 to 1.38). When only results from RCTs were pooled, the increase in risk was almost 50% (RR 1.48, 95% CI 0.84 to 2.60), but the CI was much wider. Based on the GRADE approach, the quality of the evidence was low when restricted to RCTs and very low when non-randomised studies were included. There was no difference between groups in which medication was stopped and groups in which it was continued for any secondary outcomes, but these were mainly assessed in single studies.

This is the first systematic review into a topic of major importance, as interventions of this type are currently being implemented throughout the UK and internationally, with the aim of reducing the incidence and/or severity of AKI. We have used broad inclusion criteria in many databases to capture randomised and non-randomised studies, in primary and secondary care, and for a range of AKI precipitants including intercurrent illness and planned or emergency radiological and surgical procedure. However, we have found that the published evidence was sparse and has important limitations. It is focused in hospital settings, mainly in patients undergoing coronary angiography, restricted to patients who were considered high risk for AKI and predominantly evaluates discontinuation of ACEI and ARBs. The primary definition of AKI in all studies was based on short-term changes in serum creatinine, although definitions varied across studies. The definitions of AKI used in four of the studies may have overestimated the incidence of AKI compared with the currently accepted definition of AKI, which was used in only one study.[19]

Since ACEIs/ARBs reduce glomerular filtration rate but preserve tubular blood flow, a more marked short-term reduction in eGFR may be associated with lower rates of established AKI due to ongoing tubular injury.[22] Indeed, the only study[19] that examined alternate biomarker-based definitions of AKI found no effect related to drug cessation. In addition, the longer term

**Figure 3** Forest plot showing the mean difference in glomerular filtration rate at 24 hours in those who stopped medication prior to procedure compared with those who continued medication.

**Figure 4** Forest plot showing the mean difference in creatinine at 24 hours in those who stopped medication prior to procedure compared with those who continued medication.

impact of AKI in terms of the development of CKD or reductions in baseline GFR was not reported. The reduction in glomerular filtration rate caused by ACEI and ARB treatment is reversible on stopping the drug.[23] This temporary rise in GFR among patients who discontinued the drugs might have masked AKI in the studies included here, given that AKI was defined as a change in serum creatinine from a baseline measurement that was taken prior to drug withdrawal. Recognising the potential for physiological rather than pathological changes in kidney function,[22] future studies will benefit from examining later clinical outcomes including incomplete recovery from AKI (ie, failure of serum creatinine concentration to return to baseline), chronic kidney disease and all-cause mortality.

To further quantify the limitations of the studies, we conducted a formal risk of bias assessment using the most recently developed tools. This is the first review to have used the ROBINS-I tool[14] and the new Cochrane tool for randomised trials.[13] The majority of studies were small and there were some concerns regarding risk of bias in some studies, especially one of the non-randomised studies which was judged at critical risk of bias. Publication bias was not formally assessed in this review because the number of studies was too small for such an assessment to be meaningful. However, our search strategy included a variety of routes to identify unpublished studies and resulted in the inclusion of one conference abstract. Despite this, we consider the likelihood of publication bias in this area to be high.

Importantly, there are no studies which evaluate the benefits of stopping medication in the community following acute infection, and no studies that assessed discontinuation of diuretics that could exacerbate AKI, or metformin and/or sulfonylureas, which may accumulate during an episode of AKI. Only one study assessed

**Table 3** Summary of outcomes evaluated in single studies

| Outcome | Study | Effect size (95% CI) |
|---|---|---|
| Urea (24 hours) | Wolak et al[16] | MD=2.17 (−5.22 to 9.56) |
| Diastolic blood pressure (48 hours) | Wolak et al[16] | MD=0.30 (−5.01 to 5.61) |
| Systolic blood pressure (48 hours) | Wolak et al[16] | MD=−2.10 (−12.98 to 8.78) |
| Hypertensive treatment | Wolak et al[16] | RR=0.17 (0.01 to 3.69) |
| Death | Bainey et al[18] | RR=3.15 (0.13 to 78.17) |
| Myocardial infarction | Bainey et al[18] | No events |
| Stroke | Bainey et al[18] | RR=3.15 (0.13 to 78.17) |
| Congestive heart failure | Bainey et al[18] | No events |
| Rehospitalisation | Bainey et al[18] | RR=7.49 (0.38 to 146.89) |
| Interleukin 18 (IL 18) (≥120 ng/mL) | Coca et al[19] | 0.89 (0.65 to 1.23)* |
| Kidney injury molecule 1 (KIM 1) (≥1.15 ng/mL) | Coca et al[19] | 1.09 (0.82 to 1.44)* |
| Liver fatty acid binding protein (L-FABP) (≥170 ng/mL) | Coca et al[19] | 0.97 (0.73 to 1.3)* |
| Neutrophilgelatinase-associated lipocalin (NGAL) (≥120 ng/mL) | Coca et al[19] | 0.84 (0.60 to 1.16)* |

*Adjusted for sex, age, white, CKD-EPI eGFR, diabetes, hypertension, congestive heart failure, myocardial infarction, cardiac cauterisation in past 48 hours, electic surgery and type of surgery (CABG, valve, both).
MD, mean difference; RR, relative risk.

discontinuation of NSAIDs and only a very small number of patients discontinued these drugs in this study and so it was not possible to draw conclusions regarding the effects of discontinuing NSAIDs.

At present, a number of national organisations provide guidance about medication cessation, as well as many regional schemes and guidelines. The UK NICE guidance published in 2013 recommends consideration of temporarily stopping ACEI and ARBs in adults having iodinated contrast agents if they have chronic kidney disease with an eGFR <40 mL/min/1.73 m$^2$, and in adults, children and young people with diarrhoea, vomiting or sepsis.[6] In 2015, NHS Scotland and the Scottish Patient Safety Programme initiated a more wide ranging medication cessation intervention. Predominantly via community pharmacists, patients are issued with Sick-Day Rules cards, advising them to stop taking ACEIs/ARBs, NSAIDS, diuretics and metformin when they become unwell with vomiting or diarrhoea, and/or fevers sweats and shaking.[24] Under similar circumstances, guidance from the Canadian Diabetes Association Clinical Practice Guidelines for Chronic Kidney Disease in Diabetes recommends physicians and patients to withhold ACEIs, ARBs, NSAIDs, diuretics, metformin, direct renin inhibitors and sulfonylureas.[25] This guidance is based on the commonly held belief that there is an association between the use of ACEI/ARBs, diuretics and NSAIDs and the development of AKI, particularly during illness or other physiological insult. The potentially strongest source of evidence, the incidence of AKI in RCTs of ACEIs and ARBs compared with placebo is poorly described due to variable definitions or absent reporting of kidney-related adverse events.[26] A number of observational studies have demonstrated a higher risk of AKI among patients among ACEI/ARB users also taking diuretics and/or NSAIDs compared with those taking ACEIs/ARBs alone,[27–29] or with ACEI/ARB users compared with non-users during acute illness or after surgery.[30 31] As with all observational evidence, these studies carry an inherent risk of associations being due to bias and confounding, particularly confounding by indication, in which patients at higher risk of AKI are more likely to be treated with the drugs of interest, making a direct causal effect uncertain.

Only one of the studies[17] considered in this review was available at the time of development of the NICE guidance for AKI.[6] The guideline development group discuss explicitly the difficulty of issuing guidance regarding medication cessation (for ACEIs/ARBs only), despite limited evidence.[6] They felt that the available evidence for discontinuation was weak but that the 'continuing use of ACEIs/ARBs [during acute illness or exposure to iodinated contrast agents] is clearly associated with AKI. In contrast, the temporary suspension of ACEIs/ARBs for a short period seems unlikely to greatly increase the risk of cardiovascular events'.

Subsequent evidence regarding the safety of community medication cessation interventions has come from an ongoing evaluation of hospital admissions following introduction of the NHS Scotland scheme, which has shown a stabilisation or fall in hospital admissions with AKI.[24] However, a concurrent fall in heart failure admissions (which might have been expected to increase as a consequence of discontinuation of ACEIs or ARBs among patients previously stabilised on these drugs for treatment of heart failure) suggest a secular trend in hospital admissions unrelated to the introduction of the intervention, and interpretation is also limited by the absence of a control population. There remains ongoing disagreement about how the general evidence base should be interpreted to consider the balance of risks and benefits of drug-cessation interventions, particularly during acute illness.[8]

This systematic review includes five additional studies published since the NICE guidance on AKI. Our results show low-quality evidence that withdrawal of ACEI/ARBs and NSAIDs prior to coronary angiography and cardiac surgery may reduce the incidence of AKI. However, the quality, power and limited scope of these studies reduce the emphasis that can be placed on this finding and have not substantially clarified the evidence base. There is no published evidence of the impact of drug cessation interventions on AKI incidence during intercurrent illness in primary or secondary care, of other included medications (NSAIDs, diuretics, sulfonylureas, metformin) or of combinations of medications. We also found no evidence of ongoing studies of interventions on any of these topics.

The current widespread promotion of 'sick-day guidance' incurs financial and opportunity costs. While the public health impact of sick-day guidance can be evaluated through the novel data flows recently established by NHS England and the UK Renal Registry,[3] more formal controlled evaluation in the form of stepped wedge or cluster randomised trials could be applied to ensure we achieve maximal overall public health benefit.

**Author affiliations**
[1]The National Institute for Health Research Collaboration for Leadership in Applied Health Research and Care West (NIHR CLAHRC West) at University Hospitals Bristol NHS Foundation Trust, Bristol, UK
[2]School of Social and Community Medicine, University of Bristol, Bristol, UK
[3]UK Renal Registry, Bristol, UK
[4]Department of Non-communicable Disease Epidemiology, London School of Hygiene and Tropical Medicine, London, UK
[5]Centre for Primary Care, Institute of Population Health, The University of Manchester, Manchester, UK
[6]National Institute for Health Research Collaboration for Leadership in Applied Health Research and Care (NIHR CLAHRC) Greater Manchester, Centre for Primary Care, Institute of Population Health, University of Manchester, Manchester, UK
[7]Department of Renal Medicine, Freeman Hospital, Newcastle Upon Tyne Hospitals Foundation Trust, Tyne and Wear, UK

**Acknowledgements** The authors would like to thank Dr Tim Jones, NIHR CLAHRC West, for help with assessing the risk of bias in the included studies.

**Contributors** CT, TB and FC conceived the idea for the review. PW, AM, JH and LAT drafted the article with the support of FC. AR developed the search strategy. FC, LAT, TB and CT served as content experts in the field of AKI. JH served as the overall supervisor and provided input on study methodology. JS provided methodological support. AM, PW and TS undertook screening and data extraction. PW and FC performed the risk of bias assessment. PW and JS performed the GRADE assessment. All authors contributed to the interpretation of results, commented on draft manuscripts and have given their approval for publication.

**Funding** This research is supported by the National Institute for Health Research (NIHR) Collaboration for Leadership in Applied Health Research and Care (CLAHRC) West at University Hospitals Bristol NHS Foundation Trust. TB was partly funded by the NIHR CLAHRC Greater Manchester. LAT is funded by a Wellcome Trust intermediate clinical fellowship (101143/Z/13/Z).

**Disclaimer** The funders had no role in the design of the study, data collection and analysis, decision to publish or preparation of the manuscript. However, the project outlined in this article may be considered to be affiliated to the work of the NIHR CLAHRC Greater Manchester and NIHR CLAHR West. The views expressed in this article are those of the authors and not necessarily those of the NHS, NIHR or the Department of Health.

**Competing interests** None declared.

**Provenance and peer review** Not commissioned; externally peer reviewed.

**Data sharing statement** No additional data are available.

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
