## [Reviewer comments · BMJ Open]

ARTICLE DETAILS

TITLE (PROVISIONAL)	What are the risks and benefits of temporarily discontinuing medications to prevent acute kidney injury? A Systematic Review and meta-analysis
AUTHORS	Whiting, Penny; Morden, Andrew; Tomlinson, Laurie; Caskey, Fergus; Blakeman, Thomas; Tomson, Charles; Stone, Tracey; Richards, Alison; Savovic, Jelena; Horwood, Jeremy

VERSION 1 - REVIEW

REVIEWER	Roy Mathew, MD WJB Dorn VA Medial Center Columbia, SC USA
REVIEW RETURNED	07-Jun-2016

GENERAL COMMENTS	Whiting and colleagues have performed an important systematic review/meta-analysis on the risks of continuing or discontinuing RAAS inhibitors prior to surgical procedures. As mentioned, there have been no randomized or other porospective studies on the discontinuation of RAAS inhibitors during acute illnesses. They have found a weak positive association of continuing RAAS inhibitors and AKI prior to coronary angiography or surgery. They have importantly noted the limitations of the available evidence, as well as highlighting the need for additional studies. I would only question the secondary outcomes of GFR at 24h and creatinine at 24hours. These are not standard definitions for AKI nor are they relevant outcomes in regards AKI. If GFR or creatinine are used, they are usually remote outcomes. AKI should only be judged by the change in creatinine/GFR or, currently, by biomarkers of injury. Otherwise well done and timely.
---

REVIEWER	Saoussen Ftouh National Guideline Centre (formerly known as National Clinical Guideline Centre), UK
REVIEW RETURNED	10-Jun-2016

GENERAL COMMENTS	The results are clearly presented and well discussed especially in relation to the difficulties of the varying definitions of AKI. I would however, suggest that the authors consider presenting the meta-analysis of the RCTs and cohort studies separately and not to pool the data from the 2 types of study designs together. One of the
--

	RCTs had a low risk of bias and there were some concerns about the other 2 but overall they were of fairly good quality. Of the 2 cohort studies, one had a critical risk of bias and the other one moderate. Therefore, I don't think the quality of the cohort studies is good enough to pool with the RCTs for the incidence of AKI outcome. I personally would not analyse them together and don't see the usefulness of doing this. Pooling the RCT and cohort data together is also affecting the precision of the overall effect. In the GRADE evidence profile (table 3), the imprecision goes from serious when only looking at RCTs to not serious when the data from all studies are pooled together which is not entirely reflective of the data and makes it look better than it actually is.
--	---

REVIEWER	Yong Chen University of Pennsylvania, USA
REVIEW RETURNED	30-Aug-2016

GENERAL COMMENTS	In this manuscript, the authors conducted the systematic review and meta-analysis to investigate the risks and benefits of temporary discontinuation of medications (e.g., ACE inhibitors, ARBs, and NSAIDS, etc.) for reducing the risk of acute kidney injury (AKI) for those patients with high risk AKI, prior to coronary angiography and cardiac surgery. This systematic review was based on 5 published studies. The results suggested that those patients with continued medications would likely to have higher risk of AKI than those with stopping medications. The authors gave a balanced discussion on the strengths and limitations on the analysis. Major concerns:  1. As discussed by the authors, the systematic review is likely to be subjected to publication bias. The conclusion based on the evidence synthesis from the 5 studies may be questionable. Furthermore, the fact that they combine 2 cohort studies with 3 RCTs requires special attention because the effect sizes from the cohort studies may subject to bias due to issues such as unmeasured confounders, and the study population in the cohort studies may be different from the RCTs. In addition, according to the assessment of GRADE evidence shown in Table 3, the studies were affected by very serious risk of bias, and they had very low scores on the quality. I think such conclusion may be misleading due to the analysis results based on the weak and sparse evidence. 2. Potential outcome reporting bias: as shown in Figures 3 and 4, only 2 out of 5 studies have reported mean difference in GFR at 24 hours (or mean difference in creatinine at 24 hours). Such a partial reporting may be subject to the outcome reporting bias, which is quite commonly encountered. The authors should spend efforts in obtaining the effect sizes of these outcomes to reduce the impact of outcome reporting bias. 3. Page 10, lines 33-35, only these studies "Wolak (2013)" and "Rosenstock (2008)" were used to compare the mean difference in GFR at 24 hours or in creatinine at 24 hours. In particular, both
--

	studies have small sample sizes (n=64 for the study Wolak 2013, and n=220 for the study Rosenstock 2008). The total sample size is still too small to detect meaningful mean differences in GFR at 24 hours or in creatinine at 24 hours. 4. Since these 6 studies had different timing of stopping medications and starting medications after coronary angiography/ cardiac surgery, I am wondering if both the time of development of AKI and the time of stopping medications are potential confounding factors for the risk of AKI? 5. The primary and secondary outcome measures (e.g., scale) should be described in details in the Section of Methods. 6. The structure of the manuscript can be improved if the subtitles in Results Section are provided. 7. Page 9, lines 20-24, when the data are considered as highly homogenous, it may suggest to use the “fixed effect meta-analysis” instead of “random effects meta-analysis”. 8. It would be helpful to provide a table of acronyms, for example, “NR”, and also the abbreviation of “RR”. 9. The authors should make the data available in the appendix for the reproducibility.
--	--

VERSION 1 – AUTHOR RESPONSE

Reviewer: 1, Roy Mathew, MD

Comment

I would only question the secondary outcomes of GFR at 24h and creatinine at 24hours. These are not standard definitions for AKI nor are they relevant outcomes in regards AKI. If GFR or creatinine are used, they are usually remote outcomes. AKI should only be judged by the change in creatinine/GFR or, currently, by biomarkers of injury.

Response

We are limited in the outcomes that we can report by those that are reported in the primary studies. These outcomes were reported by two of the primary studies and so we considered them relevant to include in our review. We are not considering these as measures of AKI simply as differences in GFR and creatinine as reported in the studies.

We have changed the secondary outcomes specified in the abstract to “EGFR and creatinine post AKI”.

We hope that this helps to clarify that we were interested in these measures at any time point reported in the primary studies rather than specifically at the 24 hour time point.

Reviewer: 2, Saoussen Ftouh

Comment

The results are clearly presented and well discussed especially in relation to the difficulties of the varying definitions of AKI.

I would however, suggest that the authors consider presenting the meta-analysis of the RCTs and cohort studies separately and not to pool the data from the 2 types of study designs together. One of the RCTs had a low risk of bias and there were some concerns about the other 2 but overall they were of fairly good quality. Of the 2 cohort studies, one had a critical risk of bias and the other one moderate.

Therefore, I don't think the quality of the cohort studies is good enough to pool with the RCTs for the incidence of AKI outcome. I personally would not analyse them together and don't see the usefulness of doing this.

Pooling the RCT and cohort data together is also affecting the precision of the overall effect. In the GRADE evidence profile (table 3), the imprecision goes from serious when only looking at RCTs to not serious when the data from all studies are pooled together which is not entirely reflective of the data and makes it look better than it actually is.

Response

Thank you.

We already present the results stratified according to study design – these are available in figure 2 (forest plot). The narrative description of results in the text and abstract and table 3 (GRADE assessment) presents the overall pooled results and the results of the pooled analysis restricted to RCTs.

We agree that there is debate regarding whether and when it is appropriate to combine randomised and non-randomised studies in a single meta-analysis. Guidance suggests that studies should only be

combined if they are similar in terms of population, intervention and outcome. We think this is the case for these studies. Meta-analysis should first be stratified based on study design with results from randomised and non-randomised designs only pooled if the stratified estimates are sufficiently similar. We have followed this approach and consider this to be the case.

We have added the following to the methods section to clarify our approach:

“When the same outcomes were assessed in both randomized and non-randomized studies that were considered similar in terms of population and intervention, we first stratified the analysis based on study design. If summary estimates from stratified analyses were considered sufficiently similar we then produced an overall summary estimate combining data from randomized and non-randomized studies.”

We have considered not reporting the summary estimate for both RCTs and cohort studies combined but think that for this data set it is appropriate to combine these data.

The pooled analysis for the secondary outcomes only includes RCTs as these outcomes were not assessed in the cohort studies.

We are unclear regarding the comment on imprecision and GRADE. When the summary estimate includes both RCTs and cohort studies the number of events increases considerably and the confidence intervals is much narrower. It therefore seems reflective of the data to rate this analysis as not serious for imprecision.

Reviewer: 3, Yong Chen

Comment

1. As discussed by the authors, the systematic review is likely to be subjected to publication bias. The conclusion based on the evidence synthesis from the 5 studies may be questionable. Furthermore, the fact that they combine 2 cohort studies with 3 RCTs requires special attention because the effect sizes from the cohort studies may subject to bias due to issues such as unmeasured confounders, and the study population in the cohort studies may be different from the RCTs. In addition, according to the assessment of GRADE evidence shown in Table 3, the studies were affected by very serious risk of bias, and they had very low scores on the quality. I think such conclusion may be misleading due to the analysis results based on the weak and sparse evidence.

Response

We agree with the reviewers comments regarding the limitations of the evidence – these are issues that we have highlighted and discussed in our paper.

Our conclusion in the abstract and on page 14 of the discussion state that “there is low quality evidence that withdrawal of ACE Inhibitors/ARBs and NSAIDs prior to coronary angiography and cardiac surgery may reduce the incidence of AKI” and that “there is no published evidence of the impact of drug cessation interventions on AKI incidence during inter-current illness in primary or secondary care, of other included medications (NSAIDs, diuretics, sulfonylureas, metformin) or of combinations of medications”.

Our conclusions consider the quality of the evidence and we think they are sufficiently cautious to reflect the limitations of the evidence base.

Comment

2. Potential outcome reporting bias: as shown in Figures 3 and 4, only 2 out of 5 studies have reported mean difference in GFR at 24 hours (or mean difference in creatinine at 24 hours). Such a partial reporting may be subject to the outcome reporting bias, which is quite commonly encountered. The authors should spend efforts in obtaining the effect sizes of these outcomes to reduce the impact of outcome reporting bias.

Response

As pointed out by Reviewer 1, creatinine and eGFR 24 hours post AKI are not recognised outcome measures for AKI. It is therefore very unlikely that the other investigators will have collected these items.

Comment

3. Page 10, lines 33-35, only these studies “Wolak (2013)” and “Rosenstock (2008)” were used to compare the mean difference in GFR at 24 hours or in creatinine at 24 hours. In particular, both studies have small sample sizes (n=64 for the study Wolak 2013, and n=220 for the study Rosenstock 2008). The total sample size is still too small to detect meaningful mean differences in GFR at 24 hours or in creatinine at 24 hours.

Response

Although sample sizes were small, the data suggest no evidence of a difference between groups rather than lack of power to detect a difference. For example, Rosenstock reported a MD of 0.0 for creatinine at 24 hours and Wolak an MD of -0.05. We have added the following to the results section where this is discussed to highlight the imprecision:

“..although confidence intervals were wide.”

Comment

4. Since these 6 studies had different timing of stopping medications and starting medications after coronary angiography/ cardiac surgery, I am wondering if both the time of development of AKI and the time of stopping medications are potential confounding factors for the risk of AKI?

Response

We don't think this would be confounding but could have been a source of heterogeneity between studies. We would have considered timing of medication stopping/starting as potential sources of heterogeneity. However, there was no evidence of heterogeneity for our pooled analysis ($I^2=0\%$).

Comment

5. The primary and secondary outcome measures (e.g., scale) should be described in details in the Section of Methods.

Response

Thank you – we agree this was not clear in the methods section. We have included the following sentence under the section on data extraction on page 6

“The primary outcome was incidence of AKI secondary outcomes include urinary biomarkers, clinical outcomes, creatinine, eGFR, urea and blood pressure”

Comment

6. The structure of the manuscript can be improved if the subtitles in Results Section are provided.

Response

We have added subheadings to the results section.

Comment

7. Page 9, lines 20-24, when the data are considered as highly homogenous, it may suggest to use the “fixed effect meta-analysis” instead of “random effects meta-analysis”.

Response

We do not think it is appropriate to select the meta-analysis model based on whether or not results are homogeneous. Although there are differing views on when it is appropriate to use which model, generally the choice of model should depend on the sampling frame used to select studies for the analysis. We considered that our studies did not come from the same underlying population and so a random effects model is considered more appropriate.

Comment

8. It would be helpful to provide a table of acronyms, for example, "NR", and also the abbreviation of "RR".

Response

It is our understanding that BMJ Open does not include tables of acronyms. We would be happy to include one if this would be considered appropriate. We have ensured that all acronyms are spelt out in full when first used and in footnotes to tables.

Comment

9. The authors should make the data available in the appendix for the reproducibility.

Response

We included the majority of our data in figures and tables. We would be happy to include any additional data as supplementary information but are unclear exactly what data the reviewer is referring to.

VERSION 2 – REVIEW

REVIEWER	Roy Mathew WJB Dorn VAMC USA
REVIEW RETURNED	28-Sep-2016

GENERAL COMMENTS	The authors have adequately addressed all concerns. The challenge of the analysis lies in the sparse data available to analyze. The authors have acknowledged this and have addressed the concerns in the results and discussion.
---

REVIEWER	Saoussen Ftouh National Guideline Centre, UK
REVIEW RETURNED	04-Oct-2016

GENERAL COMMENTS	I am happy with the additional wording regarding stratification of the studies before pooling the results of the cohort studies and the RCTs. I agree there is debate about this but I am satisfied that the authors gave a reasonable rationale for doing this and explained it in
---

	the article. I think the overall revisions have improved the article and I have no other issues to raise. I would recommend this article for publication.
--	--